# Manganese and Lead Exposure and Early Puberty Onset in Children Living near a Ferromanganese Alloy Plant

**DOI:** 10.3390/ijerph19127158

**Published:** 2022-06-10

**Authors:** Nathália Ribeiro dos Santos, Juliana Lima Gomes Rodrigues, Matheus de Jesus Bandeira, Ana Laura dos Santos Anjos, Cecília Freitas da Silva Araújo, Luis Fernando Fernandes Adan, José Antonio Menezes-Filho

**Affiliations:** 1Laboratory of Toxicology, College of Pharmacy, Federal University of Bahia, Salvador 40170-110, Brazil; nathalias@ufba.br (N.R.d.S.); anjos.ana@ufba.br (A.L.d.S.A.); 2Graduate Program in Pharmacy, College of Pharmacy, Federal University of Bahia, Salvador 40170-110, Brazil; juuhrodrigues@icloud.com (J.L.G.R.); matheusbandeira1@hotmail.com (M.d.J.B.); 3Environmental and Public Health Program, National School of Public Health, Oswald Cruz Foundation, Rio de Janeiro 21041-210, Brazil; cecilia.araujo@ufba.br; 4Graduate Program in Medicine and Health, College of Medicine, Federal University of Bahia, Salvador 40170-110, Brazil; luisfernando@ufba.br

**Keywords:** manganese, blood lead, children, early puberty, environmental exposure

## Abstract

Manganese (Mn) and lead (Pb) have been associated with the deregulation of the neuroendocrine system, which could potentially favor the appearance of precocious puberty (PP) in environmentally exposed children. This study aims to evaluate the exposure to Mn and Pb and their potential effects in anticipating puberty in school-aged children living near a ferromanganese alloy plant in Bahia, Brazil. Toenail, occipital hair and blood samples were collected from 225 school-aged children. Tanner’s scale was used for pubertal staging. Mn in blood (MnB), toenail (MnTn) and hair (MnH) and blood lead (PbB) levels were measured by graphite furnace atomic absorption spectrometry. Puberty-related hormone concentrations were determined by chemiluminescence. The age at which girls’ breasts began to develop was inversely correlated with weight-for-age, height-for-age and BMI-for-age Z-scores (*p* < 0.05); pubarche also had similar results. Mn biomarker levels did not present differences among pubertal classification nor among children with potential PP or not. Furthermore, Mn exposure was not associated with the age of onset of sexual characteristics for either girls or boys. However, PbB levels were positively correlated with boys’ pubic hair stages (rho = 0.258; *p* = 0.009) and associated with the age of onset of girls’ pubarche (β = 0.299, 95%CI = 0.055–0.542; *p* = 0.017). Testosterone and LH concentrations were statistically higher in boys with an increased PbB (*p* = 0.09 and *p* = 0.02, respectively). Prospective studies are needed to better assess the association between exposure to Mn and Pb and the early onset of puberty.

## 1. Introduction

Several environmental contaminants have been listed as potential endocrine disruptor chemicals (EDCs). Among them, potentially toxic metals have been investigated as a possible outcome in exposed children. Exposure to manganese (Mn) and lead (Pb) are reported to affect the timing of puberty in children, including alterations in height, weight and puberty-related hormones [1,2,3,4].

Manganese is a trace element and micronutrient of fundamental importance for certain physiological functions of the human body [5]. However, excessive exposure to it, besides causing negative impacts on the central nervous system (CNS), can still cause respiratory and reproductive problems [6]. It is known that exposure to this element is related to significant damage to growth and the reproductive system in studies with rats [7,8,9], suggesting that Mn plays a critical role in these processes. Mn stimulates the auto-oxidation of dopamine in dopaminergic neurons [10], which, in turn, is able to modulate prolactin concentrations (PRL) in the body, since it is an inhibitory regulator of this hormone [11]. On the other hand, Pb is a xenobiotic, ubiquitous environmental contaminant, associated with several health effects, including in the central nervous and neuroendocrine systems. It is known that Pb exposure plays a role in the equilibrium of puberty-related hormones and thyroid hormones, which are extremely important to the pubertal development of children, and others hormones that involve the neuroendocrine system [12]. Moreover, it is not easily possible to control for Pb exposure because it occurs naturally in the environment, and it could be a cofounder for Mn exposure, as it also affects neuroendocrine parameters.

Experimental studies with laboratory animals investigating the action of Mn on the endocrine system have been reported in recent years. Lee et al. [9] investigated the onset of puberty in rats exposed to Mn through gastric gavage and observed a significant increase in concentrations of luteinizing hormone (LH) and estradiol (E_2_) in the Mn-treated group, in which a higher accumulation of Mn was also found in the brain. Studies with male rats have demonstrated the potential of Mn to induce luteinizing-hormone-releasing hormone (LHRH) secretion, contributing to the alteration of the endocrine balance in animals [13] and the acceleration of daily sperm production [8]. Pine et al. [14] carried out the first study demonstrating an increase in serum levels of LH, follicle-stimulating hormone (FSH) and E_2_ in rats exposed to Mn. Recently, Yang et al. [15] showed that Mn is able to advance and induce early puberty through a gamma-aminobutyric acid/nitric oxide pathway in immature female rats, thus suggesting that this element can stimulate the secretion of specific hormones of puberty and trigger the onset of the pubertal process.

Regarding studies with humans, there are those that address the reproductive effects of occupational exposure to Mn. Ellingsen et al. [16] evaluated men exposed to Mn in an industrial area and observed a significant increase in PRL and LH in this group when compared with controls. Niu et al. [17] evaluated welders exposed to the welding fumes and observed that the serum PRL values in those exposed were significantly higher than in the non-exposed group. Another study found significantly higher Mn levels in the blood and urine of welders than in administrative workers [18], as well as an increase in PRL and LH serum concentrations. However, a negative correlation was observed between PRL and testosterone (T), suggesting that exposure to Mn in adulthood promotes a negative impact on male reproductive function, such as impotence and loss of libido [5]. Moreover, thyroid function is related to puberty phase, which is an important target for endocrine-disrupting chemicals, including Mn exposure [19]; however, the potential puberty-related mechanisms have not yet been elucidated.

The role of Pb in the early onset of puberty has been investigated. Dearth et al. [20] conducted a study to evaluate maternal Pb levels during pregnancy and lactation exposure and its effect on puberty-related hormones and the onset of female puberty. They reported that Pb delayed the timing of puberty, and it was associated with decreased levels of insulin-like growth factor (IGF-1), LH and E_2_, suggesting that these biochemical parameters may be involved in sexual maturation. Moreover, Tomoum et al. [21] found that boys with increased blood Pb levels presented with delayed puberty, as was also observed for girls, when they evaluated 41 children in areas of high pollution in Cairo, Egypt. Additionally, FSH and LH concentrations were decreased in children, and testosterone in boys. Recently, Liu et al. [22] measured maternal bone Pb at one month postpartum, as well as children’s blood Pb levels. They showed that higher bone and blood Pb levels postpone the age of menarche. Also observed was a negative association between girls’ pubic hair growth and childhood blood Pb levels. A later age at menarche was also demonstrated by Selevan et al. [23] in African-American and Mexican-American girls that had 3.0 µg/dL blood Pb level, with delayed breast maturation as well. In other studies, an association between blood Pb level and delayed pubic hair growth and menarche (Wu et al. [24]) and an association with breast, pubic hair and menarche in South African girls (Naicker et al. [25]) were observed.

Despite the contribution of these studies with laboratory animals and with workers evaluating the possible deregulatory action of Mn and Pb on the endocrine system, there is still little evidence of its effects on children environmentally exposed to this element, especially regarding its potentiality to anticipate the onset of puberty.

The normal age of puberty onset is usually considered to be between the ages of 8 and 13 years for the general population of girls and 9 to 14 years for boys [26]. In this context, in some condition of hormonal imbalance, puberty may be triggered sooner or later than expected. Therefore, precocious puberty (PP) is characterized by the appearance of the physical and hormonal characteristics of biological maturation before 8 years in girls and before 9 years in boys [27,28].

Evaluating precocious puberty is important because of the impact it has on children in this condition. One of the main problems related to early development is the psychosocial disorders that are observed in these children, of both sexes, such as social and behavioral problems, depression, dependence on illicit substances, eating disorders and early sexuality [29]. There is also a risk that children will not reach their potential genetic height due to rapid bone maturation, which will initially result in accelerated growth but then cease with closure of the bone epiphyses. Consequently, children will have a final height shorter than expected [30]. The diagnosis of precocious puberty is also important because it has the advantage of being able to indicate a primary sign of metabolic syndrome, obesity and even insulin resistance [31]. There is evidence that, by controlling this rapid pubertal development, issues related to anxiety and sexual abuse can be reduced, while still providing a benefit in fertility and reducing the risk of breast cancer associated with early menarche [32].

Previously, we demonstrated that children living near a ferromanganese alloy plant in Bahia, Brazil had prolactin concentrations positively associated with toenail manganese (MnTn) levels, and a non-linear association between LH concentration and MnTn in boys [33]. Thus, in this study, we hypothesize that the airborne exposure to Mn and Pb can trigger the early onset of puberty in boys and girls. Therefore, this study aims to investigate the association between the environmental exposure to these potentially toxic metals and the early onset of puberty in school-aged children living near a ferromanganese alloy plant.

## 2. Materials and Methods

### 2.1. Study Design and Population

This is a cross-sectional study carried out in the municipality of Simões Filho, located approximately 30 km from Salvador, the capital city of the State of Bahia, Brazil. We have previously shown that the entire municipality is under current atmospheric pollution from the ferromanganese alloy plant located 2 km from the urban center. The plant was inaugurated in 1970 and had an annual production of SiMn and FeMn alloys of 280,000 tonnes. More detailed information on the plant emissions can be found in our previous study [34].

Detailed information on school selection was published elsewhere [34]. Briefly, 67 elementary schools are distributed into 11 sub-regions of the municipality, with an attendance of more than 15,000 children aged 6 to 14 years. We pre-selected 15 schools (22.4% of all schools) based on their spatial distribution and the number of children ranging from 7 to 12 years old for dust sampling. Children from four elementary schools in the municipality were recruited according to the schools’ location distance from the alloy plant and the Mn dust deposition rates at the schools. The objective was to obtain an exposure gradient from low to highly contaminated environments. Thereby, the exposure groups were coded as Schools A, B, C and D, with respective Mn dust deposition rates of 1797, 5152, 9700 and 21,038 μg Mn/m^2^/30 days, as previously described by Menezes-Filho et al. [34].

The calculation of the representative sample of the number of students in this age group was based on the 95% confidence interval of the angular coefficient Beta (95%CI 1.5–6.56) observed in the effect of Mn concentration on the association with the externalizing behavior of schoolchildren, adjusted for age and sex, in Santa Luzia and Cotegipe [35]. Therefore, we estimated that the total N of students in this age group was approximately 12,000. Based on this confidence interval and the 0.80 power for type I error, we would need a sample of n = 360 children. The inclusion criteria for this study were: age between 7 to 12 years; living at least one year in the municipality; and enrolled at the selected schools. Children with diagnosed neuropsychological disturbances or already under any medication that could impair the outcome assessment were excluded from the study. Meetings were carried out at the schools with teachers, directors and the children’s parents or main caregivers to explain the study objectives and methods. After that, informed consent was voluntarily signed by parents willing to participate with their respective child. The project was approved by the Research Ethics Committee of the Federal University of Bahia (No. 874.463/2014).

Questionnaires were applied to parents or main caregivers for collecting general socio-demographic data, which were previously reported [33]. Weight and height measurements were described in that same study and were also used to calculate height-for-age, body mass index-for-age and weight-for-age Z-scores, using the software Anthroplus by WHO [36].

### 2.2. Sample Collection, Biomarker Determinations and Hormone Analysis

Hair and toenail sample collections were performed according to the methods previously described [37]. Before collection, toenail polish was removed using an appropriate acetone water solution. Hair and toenail samples were cleansed with 1% (*v*/*v*) nonionic detergent solution (Merck^®^, Triton X-100, Darmstadt, Germany), followed by 1 N nitric acid solution (JT Backer^®^, Phillipsburg, NJ, USA) and then with ultra-pure water (Milli-Q, MerckMillipore, Burlington, MA, USA). They were mineralized with concentrated nitric acid (JT Backer^®^, USA) in a closed vessel, using a microwave system (CEM^®^, Mars 6, Matthews, NC, USA). Metals were analyzed using graphite furnace atomic absorption spectrometry (GFAAS) (GTA120, AA240Z, Varian^®^, Mongrave, Australia). Standard reference material human hair IAEA-085 (International Atomic Energy Agency) was used for quality assurance purpose. The limit of detection (LOD) was 0.1 µg/g; no samples presented results below this value. Results were expressed in µg Mn/g of hair or toenails.

As described before by Dos Santos et al. [33], blood samples were collected by venipuncture of the ulnar vein, after 70% alcohol asepsis, into metal-free vacuum tubes with K_2_EDTA as an anticoagulant and also into dry tubes for serum aliquot (Vacutainer^®^ BD). For blood lead (PbB) levels, whole blood samples were diluted 1:10 with matrix modifier (phosphate buffer) before analysis by GFAAS, as described by Menezes-Filho et al. [38].

After collection and clot retraction, blood samples were centrifuged for 30 min at 14,000 rpm. Then, approximately 2.0 mL of serum was transferred into 5 mL test tubes and stored at −20 °C until analysis. The hormones PRL, LH, E2, T and TSH were analyzed by chemiluminescence using the Access equipment (Beckman Coulter^®^, Brea, CA, USA) of the Laboratory of Clinical Analysis of the Faculty of Pharmacy, Federal University of Bahia. The enzyme kits were purchased from the same equipment manufacturer and kept under refrigeration at 8 °C until the time of their use.

### 2.3. Puberty Assessment

On the same day of biological sample collection, a questionnaire to obtain information about the development of secondary sexual characteristics related to the onset of puberty was applied individually for each child, according to child’s perception of his/her own body development. In addition, the age at which their body transformation began to occur was also obtained. This questionnaire was adapted from that applied by Feibelmann [39]. For girls, the characteristics investigated were pimples, axillary hair, breast development, emergence of pubic hair and menarche. For boys: pimples, hoarse voice, axillary hair, genital development, emergence of pubic hair and first ejaculation.

The evaluation of sexual maturation was performed according to the Tanner’s Stages protocol [40], which since the 1960s has been the most used method to rank a child’s sexual maturation stages. In this context, the development of breasts (B) and pubic hair (P) in girls were evaluated, whereas in boys, the development of the genital organs (G) (testis and penis size) and pubic hair (P) were evaluated. Basically, colorful drawings depicting each sexual maturation stage were shown to the children, and they pointed out in which one they would consider themselves to fit. It is important to note that all children were interviewed individually in a quiet room by a trained interviewer of the same sex, ensuring their privacy. In order to try to ensure the reliability of the results, minimizing information bias, the interviewer approached the child in a very friendly and reliable way.

Sexual maturation stages ranged from 1 to 5 on an increasing scale of development as puberty progressed. The breasts and genitals were evaluated for their size, shape and characteristics, while the pubic hair was evaluated by quantity, characteristics and distribution. Thus, staging was performed according to the phase in which the child indicated in the panel of classical figures of Tanner study, creating a code such as B3P3 for girls and G2P3 for boys, for example.

Children were then classified in pre-pubertal and pubertal groups, and also identified with potential risk of precocious puberty (pPP) occurrence or not. For this one, all those children who had developed at least one secondary sexual characteristic (girls: thelarche or pubarche before 8 years, or menarche before 10 years; boys: development of the genitalia or pubarche before 9 years, which is the standard age for pubertal development) were classified as presenting potential precocious puberty. In those children who presented early development of pubarche, body mass index (BMI) was evaluated in order to verify whether they presented obesity or overweight, since pubarche is not very commonly the first signal of the puberty process.

### 2.4. Statistical Analysis

The distribution of children by each stage of Tanner’s pubertal development by age was reported by frequency (%). Biomarker levels were described by each stage of Tanner as median, minimum and maximum and also according to the classification of pPP. Bivariate analysis (Spearman’s correlation coefficient) was performed in order to assess correlations between the age of children and the age at onset of the secondary sexual characteristics according to hormone levels and exposure biomarkers with pubertal stages, as well as anthropometry with pubertal stages. A Chi-squared test was applied to verify whether categorical variables (sexual maturation stages per sex and low/high Mn exposure) were correlated or dependent on each other. When it was applied, one of the prerequisites for the test was not accomplished; thus, results were reported using Fisher’s Exact test. To evaluate the association between the age of puberty onset and Mn or Pb exposure, we used multivariate linear regression (MLR) models adjusted for important covariables, such as BMI and hormone concentrations. The significance level considered was *p* < 0.05. The statistical package used was SPSS version 26.

## 3. Results

### 3.1. Population General Characteristics and Sexual Maturity Staging

Initially, we recruited 245 children who fulfilled the inclusion criteria, representing 70.6% of the estimated sample size. However, sixteen children were excluded from the study because either they moved away, or we were not able to get in touch with their parents or caregivers. Four children were also excluded because they had already been diagnosed with neurological problems and were under medication. No siblings were enrolled in this study. Therefore, the final study group was composed by 113 boys and 112 girls with a mean age of 9.2 years. The mean weight, height and BMI were 33.71 kg, 137.25 cm and 18.31 kg/m^2^, respectively. None of them were statistically different between sexes (*p* > 0.05).

The mean age at which the thelarche and female pubarche were reported to have occurred was 9.0 years and menarche at 10.8 years, while for boys, the mean age of the development of the genitalia was at 9.4 years and pubarche at 9.7 years. There were no significant differences in the age of onset of these sexual characteristics among exposure groups (each elementary school).

Girls were mostly classified in both B1 (21.3%) and B2 (20.9%) stages of breast development and P2 (22.2%) pubic hair growth, and boys in stage G1 (20.9%) of the genitalia and P1 (25.3%) of pubic hair. Table 1 and Table 2 show the number of girls and boys according to their respective sexual maturation stages. No children were classified as stage 5 on Tanner’s puberty scale.

In general, approximately 64% (N = 143) of the children were classified as at puberty, among them 79 girls (55.2%) and 64 boys (44.8%), while 36% (N = 80) were in the prepubescent stage, of which 32 were girls (40%) and 48 were boys (60%). Missing data were due to some children not being available to undergo the puberty staging protocol.

Other characteristics, such as pimples, axillary hair, first ejaculation and hoarse voice, were not statistically significant with any variables tested (*p* > 0.05).

### 3.2. Pubertal Status and Anthropometry

The age at which girls’ breasts began to develop was negatively correlated with weight-for-age, height-for-age and BMI-for-age Z-scores, and pubarche also had an inverse relationship with weight-for-age and height-for-age Z-scores (*p* < 0.05) (Table 3). No correlation was observed between anthropometry and menarche (*p* > 0.05) (data not shown). Moreover, for boys, no correlation was found between the age of development of genitalia or pubic age and the evaluated Z-scores (*p* > 0.05).

### 3.3. Hormones and Pubertal Status

LH, E_2_ and T concentrations were more elevated in pubertal than in prepubertal children. Thus, higher values were found in older ones and presented more advanced stages of sexual maturation as well. For girls, LH levels were significantly more elevated at M3 and P3 (medians of 3.19 mUI/mL and 3.19 mUI/mL, respectively) compared to M1 and P1 (0.25 mUI/mL and 0.28 mUI/mL, respectively) (*p* < 0.001, for both breast and pubic hair stages). E2 levels followed the same pattern: M3 and P4 (60.0 pg/mL and 52.0 pg/mL, respectively) had significantly higher levels than at stages M1 and P1 (both with 10.0 pg/mL) (*p* < 0.001, for both breast and pubic hair stages). Only one girl was classified at stage M4. That is why this stage was not taken into account for this analysis; however, her LH and E2 levels were 2.01 mUI/mL and 52.0 pg/mL, respectively. For boys, only T levels presented a statistical difference between sexual maturation stages. Median T concentrations were higher at G4 and P4 (0.37 ng/mL and 0.94 ng/mL, respectively) when compared to stages G1 and P1 (0.1 ng/mL and 0.05 ng/mL, and *p* = 0.032 and *p* < 0.001, respectively). TSH concentration did not have any significant difference between sexual maturation stages or in those children classified as pPP or not (*p* > 0.05).

No statistical difference was observed in hormone concentrations among children classified with high levels of MnTn or not (*p* > 0.05). We stratified PbB levels into values lower than or above the recently updated cut-off from the CDC [41], that is, 3.5 µg/dL. Testosterone and LH concentrations were statistically different between the categorized variables of PbB (*p* = 0.09 and *p* = 0.02, respectively). No difference in hormone levels was observed according to pPP classification (*p* > 0.05) between boys and girls (data not shown).

### 3.4. Biomarkers of Exposure and Pubertal Status

Biomarker concentrations by sex and exposure group (school) were reported in a previous study [22]. Altogether, the median values (minimum and maximum) for girls were MnH 0.68 µg/g (0.16–5.85), MnTn 0.80 µg/g (0.15–3.24), MnB 9.94 µg/L (2.49–40.43) and PbB 1.00 µg/dL (0.3–6.6); and for boys, the values for these biomarkers were MnH 0.82 µg/g (0.28–8.79), MnTn 0.94 µg/g (0.15–13.3), MnB 8.19 µg/L (1.51–35.33) and PbB 1.20 µg/dL (0.3–15.6). In summary, the median biomarker levels of MnH, MnTn, MnB and PbB for children at school A (lower exposure) were: 0.50 µg/g (0.16–3.26); 0.54 µg/g (0.15–4.40); 11.09 µg/L (1.51–40.43); 0.60 µg/dL (0.3−5.0), respectively. In addition, for children at school D (higher exposure) the median of these biomarkers were: 0.85 (0.18–5.85); 1.88 µg/g (0.24–13.30); 8.68 µg/L (3.03–25.61); 1.45 µg/dL (0.4–5.6), respectively (detailed information described in [33]). The median levels of MnH, MnTn, MnB and PbB did not present differences among children according to their prepubertal or pubertal classification (Tanner stages of sexual maturation) (*p* > 0.05) or among children with pPP or not (Table 4). When analyzing by sex, this difference was also not identified. No correlation was found between these biomarkers and Tanner´s sexual maturation stages for either girls or boys.

However, PbB levels were positively associated with the pubertal stages in boys (rho = 0.258, *p* = 0.009) (Table 3). Significantly higher PbB levels were observed only at the P4 stage (*p* = 0.047), with a median of 2.80 μg/dL and ranging from 1.6 to 4.4 μg/dL, when compared to the P1 level (1.15 μg/dL, 0.3–4.1 μg/dL) (Figure 1).

Considering the age at which secondary sexual characteristics appeared, we found a positive relationship between PbB levels and the age of thelarche and female pubarche (rho = 0.366, *p* = 0.005; rho = 0.311, *p* = 0.011, respectively) and age of genitalia and male pubarche (rho = 0.411, *p* = 0.002; rho = 0.297, *p* = 0.045) (Table 3). The age at which the first menstrual period occurred was not correlated with the exposure biomarker levels. None of the Mn biomarkers were found to be statistically significant in relation to Tanner’s sexual maturation stages (*p* > 0.05; data not shown).

We employed the cut-off value of 1 µg/g according to the MnH levels of the Brazilian reference population [28] to test whether excessive exposure to this element is related to the parameters of pubertal evaluation. In this context, girls with lower MnTn levels had either Tanner 1 or 2 stages of breast and pubic hair maturation, respectively. In total, 49% of children with lower MnTn levels had Tanner 1, and 41.5% had Tanner 2 breast development. Those with higher MnTn levels had Tanner 1 (40.0%) and Tanner 2 (46.7%) stages of breast development (χ^2^ = 0.826). For pubic hair staging, girls with low MnTn levels were classified as Tanner 1 (40.0%) and Tanner 2 (46.2%), while 50.0% had Tanner 2 in the group with higher MnTn levels (χ^2^ = 1.209). Similarly, boys with low MnTn levels had stages G1 (48.9%) and P1 (46.7%), for both genitalia growth and pubic hair development, while in the high MnTn levels, they were classified as G1 and G2 (both 44.4%) and P1 (60.0%) (χ^2^ = 1.540 and 3.762, respectively). The Fisher’s Exact test showed that there was no association between girls’ pubertal maturation stages and MnTn exposure levels (χ^2^ = 2.782, *p* = 0.409, for breasts staging; χ^2^ = 1.857, *p* = 0.519, for pubic hair staging). Moreover, for boys, the same results were observed for both genital and pubic hair staging in relation to MnTn exposure levels (χ^2^ = 1.543, *p* = 0.658 and χ^2^ = 3.990, *p* = 0.256, respectively).

The data in Table 5 present the summary of MLR modeling, which shows that it was not possible to observe an association between MnTn and the age of onset of sexual maturation characteristics. Estradiol concentrations were positively associated with the onset age of puberty in girls (thelarche, *p* = 0.044; pubarche, *p* = 0.021), emphasizing what we have already observed in this study. When we considered the PbB level in the model, it was positively associated with the age of onset of pubarche in girls (*p* = 0.017), but it was not associated with any age of onset of puberty in boys (*p* > 0.05), after adjusting for other important covariables such as BMI and hormonal concentrations. We also carried out MLR modeling considering the exposure groups (schools); however, no relevant associations were detected (data not shown).

## 4. Discussion

This study investigated whether environmental exposure to Mn and Pb of school-aged children living near a ferromanganese alloy plant in Bahia could be associated with the early onset of puberty. Our previous study showed that these children are excessively exposed to airborne Mn but present background exposure to Pb [22,23,25]. This study found that Mn biomarkers of exposure did not significantly affect sexual maturation stages or the puberty classification (pPP or not), for either girls or boys. However, PbB levels were positively associated with pubic hair maturation in boys.

### 4.1. Sexual Maturity Staging

In the present study, the average age of onset of breasts was similar to that found in the study by Cabrera et al. [42] (9 years versus 9.7 years) performed with 610 American girls. Another study, with 1155 American girls, found that most of them entered puberty with thelarche and pubarche at the age of 10.2 years. Khadgawat et al. [43] evaluated 2010 girls aged between 6 and 17 years in India and reported the mean age for thelarche at 10.8 years and pubarche at 11 years. It was observed in the current study that, even though the girls started their physical transformation a little earlier than the children in the cited studies, the pubertal process fell within the expected age range for girls.

Boys presented the mean age for the beginning of the development of the genitalia and pubarche at 9.5 years, an age similar to that found by Tomova et al. [44], who evaluated 6200 boys, aged 10 to 19 years, and demonstrated significant growth of testicular volume at 10 years. For pubarche, the authors reported the age of onset 2.5 years later than that of our study, but that these boys still began their process of sexual maturation within the appropriate age range.

Contrasting with what is observed in the literature, we found that most cases of anticipation of secondary sexual characteristics were observed in boys. Nevertheless, the literature shows that a higher index of such traits is more frequently reported in girls [44,45]. We conjecture that boys with the desire to exalt their pubertal development reported more sexual development characteristics, configuring a bias of information. Despite this, puberty variants are expected to develop normally as the child’s ideal maturity rate is reached [46].

### 4.2. Anthropometry and Pubertal Status

Assessment of the child’s nutritional status is important because it has a direct influence on his/her growth and development. We found that the higher the values of nutritional indexes (weight-for-age, height-for-age and BMI-for-age Z-scores), i.e., the heavier the girls were, the earlier they developed secondary sexual characteristics. On the other hand, in boys, we did not observe any relation with these nutritional indexes. These findings corroborate with the study investigating the influence of obesity on early sexual maturation in boys and girls aged from 8 to 14 years [45]. The authors found that girls with early maturation were heavier when compared to boys. This puberty–obesity relationship is, in fact, better observed in girls. In the 1970s, Frisch hypothesized that “heavier girls mature earlier” because of localized fat in the subcutaneous tissue that acts as an activator of hormonal glands, stimulating the synthesis and release of puberty-related hormones [46].

### 4.3. Hormones and Puberty

Considering puberty as a simultaneous process of growth and development in which physical transformations stem from hormonal changes, assessing the relationship between the child’s age and hormone levels becomes important. Thus, the present study corroborates with data from the literature that observed that the levels of LH, E_2_ [47] and T [48] increase as children enter puberty and with the stages of sexual maturation. It is also worth mentioning that the anthropometric parameters follow the same pattern of change with increasing hormonal values as weight and height [49].

In general, in children classified as pubertal, E_2_, T and LH levels were more elevated, as expected, in those who had already started the process of sexual maturation. Similarly, puberty-related hormone secretion in prepubertal children or low-grade puberty had lower values or none of these hormones [50,51]. This is due to the physiology inherent in the human body at this stage, and therefore, it cannot suggest any influence of Mn or Pb on pubertal development.

Along with the development of pubertal characteristics, the group with higher Pb levels had also higher LH and T concentrations. It is known that Pb crosses the blood–brain barrier and can disrupt the hypothalamic–pituitary axis and increase the gonadotropin-releasing hormone [51,52], increasing the secretion of LH. It also can cross the blood–testis barrier, enabling it to alter testicular functions, such as sperm and testosterone production. However, our results are not in accordance with the ones found in the literature. Some studies have already reported significant impairment of testis functions, and that testosterone levels were reduced in boys with high Pb levels (≥10 µg/dL) [21]. Khalaf et al. [4] conducted a case–control study with male children aged 15 years in three different regions of an industrialized zone and observed that children near the plant site had higher PbB (mean of 6.38 µg/dL), while the reference group (located more than 20 km from the plant site) had a mean PbB level of 1.85 µg/dL. Children with higher levels of PbB had lower levels of testosterone (0.83 ng/mL), and the ones with lower levels of PbB had higher levels of T (1.61 ng/mL). Ronis and collaborators [53] exposed pregnant rats to lead acetate in drinking water at levels of 0.0%, 0.15% or 0.45% (*w*/*v*) and observed decreased T levels at higher doses in male pups during puberty. On the other hand, Chen et al. [54] reported an association between blood lead levels and reproductive hormone levels in men and postmenopausal women, observing T levels positively associated with Pb levels. There is still limited information concerning Pb exposure and reproductive hormones with regard to pubertal timing.

### 4.4. Biomarkers, Pubertal Status and Potential Precocious Puberty Relationship

The Mn biomarker levels of this current study were lower than the values we have already previously discussed [35,55,56,57], when we observed a median level 10-fold higher for MnH. Probably, this result could be due to the different levels of exposure of the groups, the distance of the community from the Mn transformation plant, and an emission pattern that could have diminished over the years; it could also be due to the one additional step with nitric acid during the washing procedure. However, Lucchini et al. [58] used a similar cleaning procedure and still found higher MnH levels. On the other hand, MnH levels in our children were higher than other studies where they were exposed to different types of Mn exposure [58,59], but lower than children exposed to Mn in drinking water in Canada (1.95 µg/g) [60]. MnB levels were similar to other studies, such as in Italy [58].

It has already been reported that Mn, as an environmental contaminant that is also an essential element for metabolic, physiological, growth, reproduction and neurodevelopmental processes, can further play an important role in pubertal function. Pine et al. [14] assessed the ability of Mn to stimulate the actions of the hypothalamus, which were possibly associated with the onset of puberty, and found increased serum of LH released from the pituitary, as well as the luteinizing-hormone-releasing hormone (LHRH). Lee et al. [8] also reported a chronic exposure to Mn and its relation affecting puberty-related events, demonstrating that Mn is capable of activating LHRH in male rats, which is responsible for stimulating the secretion of the sexual gonads.

Furthermore, Mn has also been demonstrated to be able to activate a specific gene that regulates the hypothalamic gonadotropin-releasing hormone (GnRH), suggesting another possible mechanism by which this element is involved in the induction of precocious puberty [61]. Other studies also investigated Mn exposure and its potential to induce precocious puberty, taking into account the role of the gamma-aminobutyric acid (GABA) receptor and the nitric oxide pathway [15], as well as the increasing hormone receptor expression of mammary epithelial cells [62].

Although there are still very few investigations into the effects of excessive exposure to Mn on pubertal development, the results of studies with animals and humans suggest that this element may stimulate the emergence of sexual characteristics that define puberty [8,14,18]. However, our study was not able to detect any association between environmental Mn exposure and the onset of precocious puberty.

On the other hand, our results suggest that Pb exposure may also have a role in the onset of precocious puberty. Pb is ubiquitous in the environment, and its neurotoxic effects have been thoroughly described in the scientific literature. Studies suggest a trend of precocious puberty onset in children associated with Pb exposure [63,64]. Lead exposure during central nervous system development and sexual differentiation can affect the normal course of puberty. Studies have reported that Pb exposure could influence pubertal development, altering the hypothalamic–pituitary–gonadal axis [20,53], spermatogenesis and reproductive functions, including puberty-related hormones such as estradiol, luteinizing hormone and follicle-stimulating hormone [49,50]. Therefore, it can act as an endocrine disruptor chemical and promote early or delayed puberty in this context. Other studies reported delayed puberty in children exposed to Pb [23,25] and in studies with animal models [20,65]. Despite that, the study performed with children from New York City did not find any association between blood Pb levels and breast or pubic hair development in girls [66]. The ones that reported delayed puberty in children presented higher Pb levels. These prior studies were not in accordance with the results found in this current investigation.

The mechanism of altered pubertal onset due to Pb exposure is still unclear, but it may be related to an alteration in the activation of GnRH [52]. Additionally, it could also involve insulin-like growth factor 1 (IGF-1), considering the inhibition of the hypothalamic–pituitary–growth axis, testosterone and other hormones responsible for pubertal development [67].

Furthermore, our data showed that PbB levels were positively associated with pubertal stages in boys. It was also positively associated with the age of pubarche in girls, after adjusting for BMI and hormonal concentrations, which, as far we know, has previously never been observed. Median PbB levels observed in the current study were similar to some studies that investigated delayed or precocious puberty in children [23,68] and lower than our previous study at the same metallurgical site. However, they did not evaluate puberty status in relation to Pb exposure [38]. A recent study demonstrated a retardation in the growth and development of both genitalia and pubarche staging in groups exposed to Pb living near an industrial area [4], who had higher PbB levels higher than the levels observed in this current study. Moreover, Liu et al. [22] assessed whether early Pb exposure was related to pubertal development in Mexico and found delayed pubertal development in girls. To our knowledge, the only study that observed younger sexual maturation in girls associated with elevated blood Pb levels was the one carried out by Choi [68]. In this recent study, the author considered the age at menarche as the major parameter used to diagnose precocious puberty. The author used the data from the Korean National Health and Nutrition Examination Survey (KNHANES) to assess the relationship between blood heavy metal levels and age at menarche in Korean girls. However, he did not consider the sexual maturation stages of puberty as we did in the current study, which is more reliable and considered as a gold standard for this diagnosis (Tanner’s stages of sexual maturation scale). These differences in findings could be explained by differences in the study population, the diversity of social environments and Pb exposure. Regarding this context, it is still inconclusive whether Pb has a role in pubertal onset period. Thus, more detailed epidemiological studies need to be carried out to try to elucidate the specific effect of Pb on pubertal onset.

Nevertheless, PbB levels were higher in older children. Therefore, we do not know whether this association is due to collinearity with age [69]. That is, older boys have higher PbB levels, probably because they are more exposed to outdoor environments, having more contact with contaminated soil or traffic emissions. On the other hand, a recent study revealed decreasing concentrations of PbB levels with the increase in age of children in eastern Iran [70]; younger children absorb Pb more easily than older ones and adults, posing more risk to their lives. Due to the cross-sectional nature of this study, we cannot ascertain the temporality issue in its causal association with elevated PbB levels and puberty indicator changes.

### 4.5. Strengths and Limitations

To the best of our knowledge, this is the first study to approach environmental Mn and Pb exposure and their effects on early puberty, bringing to discussion aspects related to secondary sexual characteristics, Tanner sexual staging, anthropometric measurements and their relation to four different biomarkers of exposure to metals (MnH, MnTn, MnB and PbB), which reflect different time-windows of exposure.

Limitations of the current study include, firstly, the lack of a physical evaluation of the children through medical examination by a pediatrician to examine the breasts, observe the pubic hair and use a suitable instrument to measure the testicular volume (orchidometer) in order to confirm with greater accuracy the stage of sexual maturation in which the children were ranked. Additionally, as the interviews were carried out inside the schools, parents/caregivers were not present at that moment, which could be pointed out as another difficulty for children’s sexual evaluation. Moreover, due to budgetary limitations, it was not possible to perform specific tests, such as radiography, to establish the exact bone age of each child. This may have contributed to some degree of bias of classification regarding pubertal staging. Secondly, the occurrence of very few cases of potential early puberty may have contributed to the failure to observe any relationship between Mn levels and precocious puberty. Thirdly, we used a different method to clean hair and toenail samples, using an additional step of acid washing, and it may have contributed to finding lower biomarker levels. Finally, the cross-sectional design limited the establishment of a causal relationship due to the temporality issue.

## 5. Conclusions

In the present study, it was not possible to observe that an excessive manganese exposure was associated with early pubertal development, although blood lead levels were positively associated with pubic hair development in boys, suggesting a role in early pubertal onset. Furthermore, hormone levels did not present any difference considering low or high Mn exposure. In addition, we could also observe that overweight girls, but not boys, were considered more likely to present potential early development of pubertal characteristics. Finally, this study was the first to evaluate in a comprehensive approach the potential onset of early puberty due to environmental exposure to potentially toxic metals using several biomarkers. Despite our findings, the hypothesis raised by this current study—that children environmentally exposed to manganese may potentially be susceptible to precocious puberty—should be further investigated in a more elaborate prospective epidemiological study.

## Figures and Tables

**Figure 1 ijerph-19-07158-f001:**
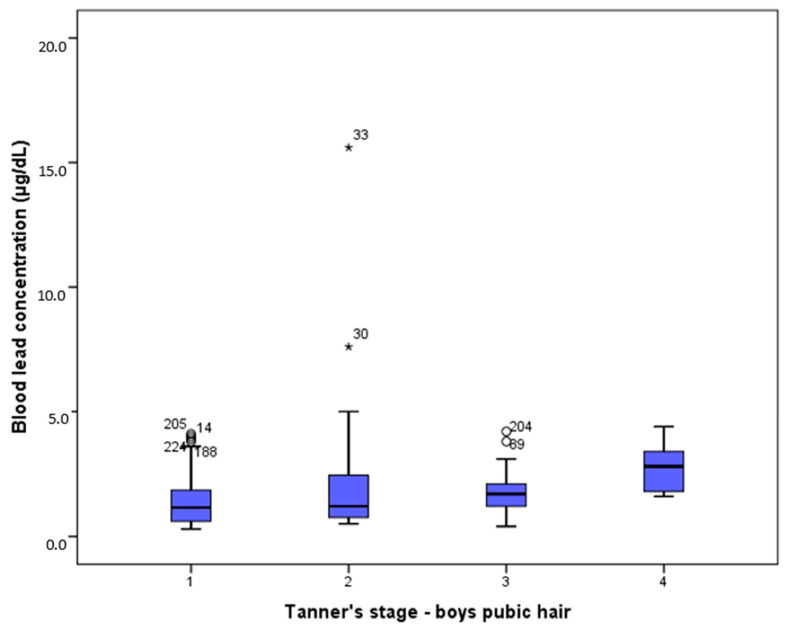
Box plot of PbB levels according to stages of pubic hair maturation in boys, according to Tanner. * denotes outliers.

**Table 1 ijerph-19-07158-t001:** Frequency (%) of girls according to Tanner’s sexual maturation stages and age.

**T. Breast**	**7 Years**	**8 Years**	**9 Years**	**10 Years**	**11 Years**	**12 Years**
**M1**	18 (37.5)	12 (25.0)	12 (25.0)	5 (10.4)	1 (2.1)	
**M2**		9 (19.1)	12 (25.5)	17 (36.2)	6 (12.8)	3 (6.4)
**M3**				1 (11.1)	6 (66.7)	2 (22.2)
**M4**						1 (100.0)
**T. Pubic Hair**	**7 Years**	**8 Years**	**9 Years**	**10 Years**	**11 Years**	**12 Years**
**P1**	13 (32.5)	16 (40.0)	5 (12.5)	5 (12.5)		1 (2.5)
**P2**	5 (10.0)	5 (10.0)	18 (36.0)	14 (28.0)	7 (14.0)	1 (2.0)
**P3**			1 (9.1)	2 (18.2)	5 (45.5)	3 (27.3)
**P4**				2 (50.0)	1 (25.0)	1 (25.0)

Note: N Tanner breast = 105; N Tanner pubic hair = 105.

**Table 2 ijerph-19-07158-t002:** Frequency (%) of boys according to Tanner’s sexual maturation stages and age.

**T. Genital**	**7 Years**	**8 Years**	**9 Years**	**10 Years**	**11 Years**	**12 Years**
**G1**	13 (27.7)	13 (27.7)	9 (19.1)	6 (12.8)	3 (6.4)	3 (6.4)
**G2**	4 (9.8)	3 (7.3)	17 (41.5)	6 (14.6)	6 (14.6)	5 (12.2)
**G3**		1 (8.3)		3 (25.0)	6 (50.0)	2 (16.7)
**G4**			1 (33.3)		1 (33.3)	1 (33.3)
**T. Pubic Hair**	**7 Years**	**8 Years**	**9 Years**	**10 Years**	**11 Years**	**12 Years**
**P1**	17 (29.8)	14 (24.6)	15 (26.3)	6 (10.5)	2 (3.5)	3 (5.3)
**P2**		2 (7.4)	8 (29.6)	6 (22.2)	6 (22.2)	5 (18.5)
**P3**		1 (7.7)	3 (23.1)	2 (15.4)	6 (46.2)	1 (7.7)
**P4**			1 (16.7)	1 (16.7)	2 (33.3)	2 (33.3)

Note: N Tanner genital = 103; N Tanner pubic hair = 103.

**Table 3 ijerph-19-07158-t003:** Spearman correlation coefficients among anthropometric parameters, exposure biomarkers and onset age of Tanner’s characteristics for girls and boys.

		1	2	3	4	5	6	7	8	9	10	11
**1 W/A Z-score**	**rho**		**0.707 ****	**0.818 ****	0.004	0.020	0.047	−0.069	−0.322	**−0.365 ***	−0.244	0.024
	***p*-value**		**0.000**	**0.000**	0.963	0.825	0.584	0.422	0.095	**0.026**	0.274	0.895
**2 H/A Z-score**	**rho**			**0.316 ****	−0.062	0.103	0.065	**−0.164 ***	−0.169	−0.235	−0.186	−0.106
	***p*-value**			**0.000**	0.390	0.161	0.352	**0.019**	0.206	0.057	0.221	0.439
**3 BMI/A Z-score**	**rho**				0.077	−0.010	−0.021	0.070	−0.057	−0.230	**−0.302 ***	0.067
	***p*-value**				0.284	0.891	0.768	0.319	0.671	0.064	**0.044**	0.625
**4 MnH (µg/g)**	**rho**					**0.388 ****	−0.114	0.185 **	0.083	0.028	−0.101	−0.071
	***p*-value**					**0.000**	0.104	0.008	0.529	0.821	0.510	0.611
**5 MnTn (µg/g)**	**rho**						−0.119	**0.244 ****	0.121	0.141	−0.109	−0.030
	***p*-value**						0.099	**0.001**	0.386	0.275	0.490	0.837
**6 MnB (µg/L)**	**rho**							−0.079	0.081	−0.112	0.044	−0.234
	***p*-value**							0.245	0.544	0.376	0.773	0.080
**7 PbB (µg/dL)**	**rho**								0.103	0.210	0.136	0.221
	***p*-value**								0.437	0.090	0.367	0.099
**8 Age of thelarche**	**rho**									0.183		
	***p*-value**									0.207		
**9 Age of pubarche—girls**	**rho**											
	***p*-value**											
**10 Age of pubarche—boys**	**rho**											**0.536 ****
	***p*-value**											**0.001**
**11 Age of genital—boys**	**rho**											
	***p*-value**											

* *p* < 0.05; ** *p* < 0.01; W/A Z-score: weight-for-age Z-score; H/A Z-score: height-for-age Z-score; BMI/A Z-score: BMI-for-age Z-score. Onset age of each sexual maturation characteristic reported in years. Bold characters are to emphasize significant correlations.

**Table 4 ijerph-19-07158-t004:** Children’s biomarkers of exposure expressed in median (minimum–maximum) by each Tanner’s sexual maturation stage and potential precocious puberty classification.

	MnH (µg/g)	MnTn (µg/g)	MnB (µg/L)	PbB (µg/dL)
	**Girls’ Tanner sexual stage**			
**M1**	0.64 (0.18–3.54)	0.64 (0.15–3.22)	10.54 (4.02–40.43)	1.00 (0.3–5.6)
**M2**	0.72 (0.16–5.85)	0.80 (0.22–3.24)	9.06 (2.49–25.50)	0.9 (0.3–6.6)
**M3**	0.56 (0.26–4.01)	0.85 (0.36–2.15)	10.07 (5.22–24.50)	1.3 (0.6–4.0)
**P1**	0.64 (0.33–3.54)	0.65 (0.15–3.22)	8.90 (3.00–23.41)	1.00 (0.3–3.3)
**P2**	0.66 (0.16–5.85)	0.75 (0.29–3.24)	10.78 (2.49–40.43)	0.9 (0.3–6.6)
**P3**	0.56 (0.18–4.01)	0.85 (0.36–3.22)	9.22 (3.25–24.33)	1.00 (0.4–4.0)
**P4**	0.97 (0.46–1.35)	0.77 (0.28–2.13)	6.66 (4.98–24.50)	1.4 (0.5–3.2)
	**Boys’ Tanner sexual stage**			
**G1**	0.83 (0.28–4.41)	0.84 (0.15–10.75)	7.22 (1.51–28.89)	1.00 (0.3–4.0)
**G2**	0.86 (0.29–6.85)	1.11 (0.35–13.30)	10.34 (2.82–32.40)	1.2 (0.4–15.6)
**G3**	0.66 (0.31–2.20)	0.74 (0.25–4.03)	8.23 (3.16–11.90)	1.65 (0.5–3.1)
**G4**	0.61 (0.55–0.82)	0.73 (0.59–5.10)	6.11 (4.93–17.09)	2.5 (1.8–4.4)
**P1**	0.79 (0.32–4.41)	1.07 (0.15–11.51)	9.28 (1.51–23.97)	1.0 (0.3–4.0)
**P2**	0.86 (0.28–6.85)	1.02 (0.29–13.30)	7.58 (2.56–32.40)	1.1 (0.5–15.6)
**P3**	0.74 (0.31–5.52)	0.83 (0.25–4.03)	9.21 (3.21–21.98)	1.6 (0.4–4.2)
**P4**	0.70 (0.55–0.86)	0.94 (0.59–5.10)	5.58 (3.16–17.09)	**2.5 (1.6–4.4) ***
	**Potential precocious puberty classification**			
**Yes**	0.64 (0.39–3.37)	0.73 (0.28–5.55)	10.22 (4.98–28.78)	1.00 (0.4–3.1)
**No**	0.73 (0.16–8.79)	0.84 (0.15–13.30)	8.94 (1.51–40.43)	1.2 (0.3–15.6)

* and bold characters represent *p* < 0.05, One-Way ANOVA with Tukey post hoc test. Note: Stage M4 not included because there was only one girl in this classification.

**Table 5 ijerph-19-07158-t005:** Multivariate linear regression model for Mn biomarker as predictor of early age (years) of puberty onset on children.

	Non-Standardized β Coefficients	95% Confidence Interval	*p*-Value
**Girls**			
**Onset age of thelarche**			
Intercept Y	8.891	6.366–11.416	**<0.001 ***
MnTn (µg/g)	0.203	−0.248–0.653	0.398
E_2_ (pg/mL)	0.007	0.000–0.013	**0.044 ****
PbB (µg/dL)	0.292	−0.013–0.598	0.060
**Onset age of pubarche**			
Intercept Y	8.023	6.008–10.038	**<0.001 ***
MnTn (µg/g)	0.306	−0.108–0.720	0.143
E_2_ (pg/mL)	0.007	0.001–0.014	**0.021 ****
PbB (µg/dL)	0.299	0.055–0.542	**0.017 ****
**Boys**			
**Onset age of genitalia**			
Intercept Y	9.073	8.061–10.085	**<0.001 ***
MnTn (µg/g)	0.086	−0.053–0.226	0.218
T (ng/mL)	−0.076	−1.021–0.869	0.871
PbB (µg/dL)	0.085	−0.060–0.229	0.240
**Onset age of pubarche**			
Intercept Y	8.994	8.005–9.983	**<0.001 ***
MnTn (µg/g)	0.101	−0.055–0.257	0.195
T (ng/mL)	0.032	−0.747–0.810	0.934
PbB (µg/dL)	0.030	−0.106–0.166	0.655

* *p* < 0.001; ** *p* < 0.05. Note: Models adjusted for BMI (kg/m^2^), LH (mUI/mL), TSH (mUI/mL). Bold characters are to emphasize significant *p*-values.

## Data Availability

The data and materials are available with the research team and will be made available on request.

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
