# Peer review of "Manganese and Lead Exposure and Early Puberty Onset in Children Living near a Ferromanganese Alloy Plant"

_ijerph, 2022, doi:10.3390/ijerph19127158_

Round 1
Reviewer 1 Report
All comments are in the attached document
Author Response
Thank you very much for the comments and suggestions. We are sure that all remarks were addressed aand really helped to make the manuscript more readable and straightforward.

Author Response
Thanks. We appreciate very much your comments, which helped made the manuscript clearer.
Reviewer 3 Report
Very interesting paper. No comments.
Author Response
Thanks. We appreciated very much your comments which helped to improve the manuscript.
This manuscript is a resubmission of an earlier submission. The following is a list of the peer review reports and author responses from that submission.
Round 1
Reviewer 1 Report
The research objectives have no scientific basis. Attempts to find a statistical association between metal content and puberty parameters are not scientific. If you aim to find correlations between any biochemical and physiological parameters, you will always find something. Statistical significance is not biological significance. It can be random. But any research and its results must be scientifically substantiated, and the results must be proven. I think that this work should not be sent to scientific journals.
Reviewer 2 Report
The study was designed to understand whether early life Mn exposure impacts the pubertal development. Most results reported were negative except for Pb. Since the manuscript was focused on Mn, throughout the Introduction, the authors did not provide sufficient justification on why Pb was included. Authors might want to revise the title changing it from “Mn exposure” to “Toxic metals exposure.” The authors should also include the measurements of other toxic metals in additional to Pb and Mn.
The authors claimed that “The breasts and genitals were evaluated for their size, shape and characteristics while the pubic hair by their quantity, characteristics and distribution” (page 4, section 2.21). However, under limitations of the study (page 14. Section 4.5), the authors stated that “As limitations of the present study, firstly, the lack of a physical evaluation of the children through physical examination of the breasts, observation of the pubic hair and use of a suitable instrument to measure the testicular volume (orchidometer)”. These two statements are contradictive to each other. This is a fundamental issue.
Also did any child, boy or girl in the study have siblings? Were any of their siblings also enrolled in the same study? If so, how was this issue handled in the data analysis?
Proof reading and English spelling check are required. For example: on page 5, it should be "Missing data", not "Miss data." On page 12 " ....posing more risk to their life" instead of "their half."